# Children in Need: Evidence for a Children's Cult from the Roman Temple of Omrit in Northern Israel

## Adi Erlich

The Zinman Institute of Archaeology, University of Haifa, Haifa 3498838, Israel; aerlich@research.haifa.ac.il

**Abstract:** Excavations of the Roman temple at Horvat Omrit, situated in the foothills of Mount Hermon and the Golan, yielded terracotta figurines dated from the first century BC—first century AD. Some 100 fragments of figurines portray young children standing with arms lifted up from the sides and bent at the elbow, palm turned outward. Although this group is unique in its iconography, it fits in with nearby temples in Phoenicia, where numerous figurines and statues of children were consecrated. Images of children from temples around the Mediterranean are often associated with healing cults and rites of passage. The child figurines from Omrit are examined with regard to their gesture, age, and gender, in order to reconstruct the likely cult that took place in the temple. The picture emerging from the terracottas is of family rites celebrating a crucial threshold in life, when passing from infancy to childhood at around the age of three. This is a vulnerable stage in childhood, since mortality rate among young children was very high in ancient societies, and rites were performed to protect them. These rites have further significance in terms of socialization, in introducing the infant to the family, to the cult, and to society in general.

**Keywords:** Roman temple at Omrit; Roman Phoenicia; archaeology of children; terracotta figurines; healing cults; rites of passage

## 1. Introduction

Religion in the southern Levant during the Hellenistic and Roman periods has been intensively studied over the past few decades. Research has focused mostly on the religious landscape of the region with its temples and cult places (Segal 2013; Raja 2017). Other studies have aimed to cover a certain region of the Levant (for Israel, see Ovadiah and Turnheim 2011; for Lebanon, see Aliqot 2009; for Syria, see Steinsapir 2005; Mazzilli 2018). These studies focus on two main aspects of religious life: (1) cult places, with a focus on architectural remains of temples and shrines; and (2) the deities who were venerated in them, relying on inscriptions, historical sources, statues, and reliefs. The third element in the sphere of religion, perhaps the most important—i.e., the worshippers themselves performing the cult—is often left aside because of the scarcity of evidence to illuminate this element.

Cult and rites can be deciphered from votives and offerings left at the cult place (Osborne 2004; Weinryb 2016); however, these are rare finds in Hellenistic and Roman temples of the southern Levant. Exceptional in this regard is Hellenistic Phoenicia, whose temples have been studied not only in relation to their architecture but also pertaining to reliefs, statues, terracotta figurines and other relics found in them (for a recent survey, see Bonnet 2015). Rich evidence comes from temples on the southern Lebanese coast, such as the cultic reliefs from the temple at Umm el-Amed, south of Tyre (Dunard and Duru 1962; Nitschke 2011; Annan 2013; Michelau 2015), terracottas from the shrine at Kharayeb, north of Tyre (Chéhab 1951–1954; Lancellotti 2003; Oggiano 2012, 2015; Bonnet 2015, pp. 245–50), and statuary and reliefs from the Eshmun complex near Sidon (Stucky 1993; Bonnet 2015, pp. 211–45). A good example of reconstructing cults from pottery found in a cultic compound in the vicinity of Omrit is in the seminal paper by Berlin (1999) on the archaeology of ritual in the Pan sanctuary at Banias.

Although Phoenician and Levantine cults are widely discussed in research, they are seldom discussed in relation to rites of passage. The pioneering work by Arnold van Gennep in 1909 (van Gennep 1960; English translation) on rites of passage provided a beneficial model in anthropology and archaeology. His study showed how rites related to stages in the life cycle mark the passage from one social state to another. Such rites include three main stages: separation from the old status, transitional stage of crossing a symbolic and/or physical threshold, and integration into the new status. Turner's study, based on van Gennep's fundamental work, put more emphasis on the liminal nature of rites of passage (Turner 1967), and Leach (1976, p. 77) focused on the element of separation in the process. The model of van Gennep and his successors was adopted by archaeologists to explain ritual behavior.

The main stages in the life cycle that were studied in relation to rites of passage are birth, puberty, marriage, and death. Rites related to newborns were also widely discussed (van Gennep 1960, pp. 50–53). Less attention was paid to rites in childhood, marking the transformation from baby to infant and from infant to child. This paper aims to fill this gap, by analyzing a corpus of figurines from the Roman temple at Omrit in northern Israel.

## 2. The Temples at Omrit

Horvat Omrit,[1] in whose Roman-era temple the figurines discussed in this paper were found, is situated in the foothills of Mount Hermon and the Golan (Figure 1). The site is close to a major junction of the Roman road network and lies just north of a perennial stream (Overman and Schowalter 2011, p. 1; Nelson 2011, p. 27; Mazor 2011, p. 19). Omrit is a liminal site in all senses: it operated on the seam between the Hellenistic and the Roman periods, and it lies on the border of Phoenicia, Syria, and Galilee, as well as on the border between the client kingdom of Herod the Great and the Ituraeans. As such, it served as a meeting point among cultures, which makes it an interesting test case for the study of cult and ritual.

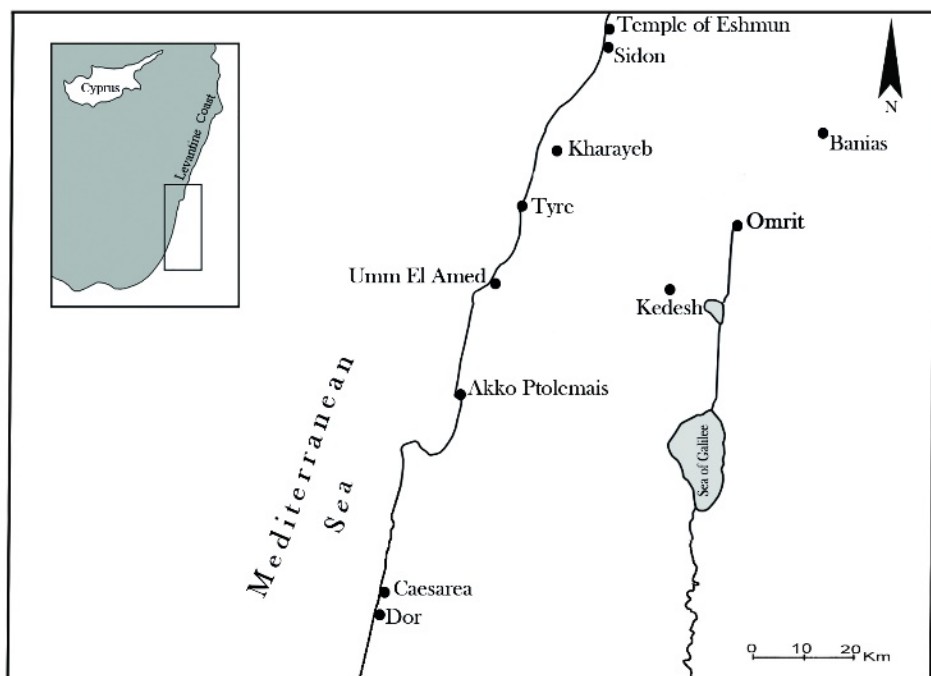

**Figure 1.** Map of the region with Omrit and other main sites mentioned in the paper (prepared by Silvia Krapiwko).

In the first century BC a shrine was erected at the site, later developed as a temple, until its final destruction in the fourth century AD. The cultic compound was composed of a temenos, altar, and three or four buildings built over time, each enveloping its predecessor

(Overman et al. 2021, pp. 5–30) (Figure 2). The first building, called by the excavators the "early shrine," had two phases, and dates to ca. 50 BC. The shrine was followed by Temple 1, founded ca. 20 BC at the time of Herod the Great. The last building, Temple 2, was constructed toward the end of the first century AD and was destroyed in the 363 AD earthquake. The buildings were lavishly adorned with wall paintings and architectural decorations (Rozenberg 2011; Nelson 2015).

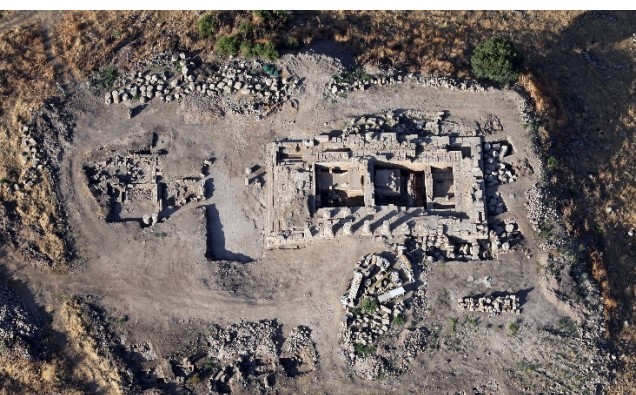

**Figure 2.** Aerial view of the Omrit temple (courtesy of the Omrit Temple Project).

The cult place at Omrit predated the nearby city, Caesarea Philippi, which was founded in 2 BC some 4 km to the northwest, near the sacred cave of Pan. Omrit was not within the borders of Caesarea Philippi, and the site seems to be an extra-urban temple. Unfortunately, there is no solid evidence for the nature of the cult at the place and the deities worshipped at the temple. It has been assumed that Temple 1 was the Augusteum erected by Herod the Great, serving for an imperial cult (for a summary of the different opinions, see Overman 2021, p. 3). An inscription found in the temple reads Aphrodite (Schowalter 2011, p. 79), and two or three figurines depict the goddess of love, alongside other figurines portraying Herakles and winged deities (Erlich 2021, pp. 145–47, 368, nos. 1–6). An altar was dedicated to Zeus (Prosch 2021), and another inscription from the late first century AD mentions the nymph Echo (Schowalter 2021). It seems that various cults took place at the site, either simultaneously or throughout time.

Other than pottery in and around the temples, findings were not numerous at the site. Among them were glass bottles, oil lamps, a few sculptural reliefs or furniture and a few inscriptions (Schowalter 2011; Rosenthal-Heginbottom 2017). It should be noted that clean and sealed contexts are scarce, because each construction phase plundered and cut through the previous one, moving around fills and finds.

Many fragments of terracotta figurines were found throughout the temple and its temenos (Erlich 2021). Most of them are related to the early shrine and Temple 1 during its lifetime, mainly the late first century BC and the first century AD. During the construction of Temple 2 in the late first century AD, the figurines were scattered. Many of them were found in the fills in and around the temple itself. The terracottas from Omrit are discussed here in the attempt to reconstruct the likely cult at this site.

## 3. The Terracotta Figurines from the Omrit Temple

About 50 figurines, half of the identifiable ones, consist of small-scale figurines, about 15 cm high, depicting gods, mortals, and animals (Erlich 2021, pp. 144–53). This group is typical of assemblages of the late Hellenistic and Early Roman periods from the Hellenistic southern Levant, and especially Phoenicia (Chéhab 1951–1954; Erlich 2009, pp. 41–60). The attribution to Phoenicia is evident, as proven by one rare type of figurine, portraying a priest holding a cultic vessel for incense or libation. This iconography is paralleled exclusively in Hellenistic Phoenicia—-in terracotta figurines from Kharayeb and on the cult reliefs from Umm El-Amed (Figure 3, left; Michelau 2015). At Omrit, only a small fragment

of the priest's arm holding the cultic vessel remained (Figure 3, right). Likewise, many of the lamps of the early phases belong to Phoenicia (Rosenthal-Heginbottom 2017, p. 450).

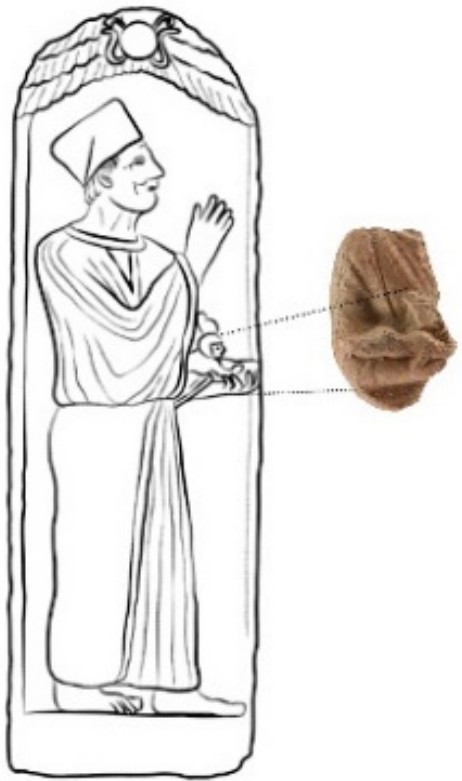

**Figure 3.** (**Left**) Drawing of a stele with a relief of a priest, from Umm El-Amed (courtesy of Henrike Michelau); (**Right**) figurine fragment of a man holding a ritual cultic vessel from Omrit (photo by Gaby Laron).

About 100 fragments, of which 50 were identifiable, comprise a separate group of large-scale human figures (Erlich 2021, pp. 153–60). Since all the figurines were found in fragments, their full height was estimated according to natural body proportions, relying mostly on the palms (10% of body height). Head–body ratio was also calculated, although this proportion varies between children and adults. Since the figurines portray children (see below), a ratio of 1:5 or 1:6 was used. Both calculations of palm to body and head to body yielded similar results, which varies between 30 and 50 cm tall. This large-scale group forms the focus of this paper.

### 3.1. The Large-Scale Terracottas from Omrit
### 3.1.1. Description of the Large-Scale Group

The large-scale series of terracottas depict standing children of different ages, with uplifted arms and outturned palms. They are made in buff to light orange fabric. Ten fragments of limbs were sent for petrographic analysis, which revealed that the clay came from the central Phoenician coast, in the Tyre area (possibly for a local workshop, see below).

Ten fragments of heads were preserved (Figure 4), one portraying a boy, and the rest depicting toddlers, with finely retouched faces. The young age of most figures, about 2–3 years, is shown in their fleshy faces. Arms (Figure 5), include both left (10 fragments) and right (6 fragments), so either each child raises a different arm, or more likely, they lift both arms. Many more limb fragments were found, without the palm. The palms, which vary in length from 3 to 5.5 cm, are depicted open and probably outturned, with the thumb set aside. In fragments of forearms or wrists, there are signs of spiral or plain bracelets, similar to those portrayed on Roman mummies from Egypt (Walker and Bierbrier 1997,

pp. 174–76). Some of the hands are rigid and schematic while others are softer and more naturalistic.

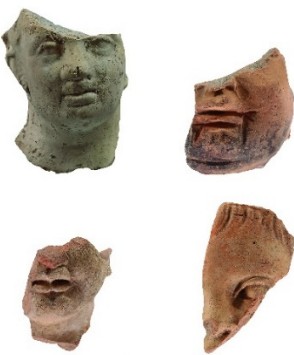

**Figure 4.** Terracottas depicting heads from Omrit (photo by Gaby Laron).

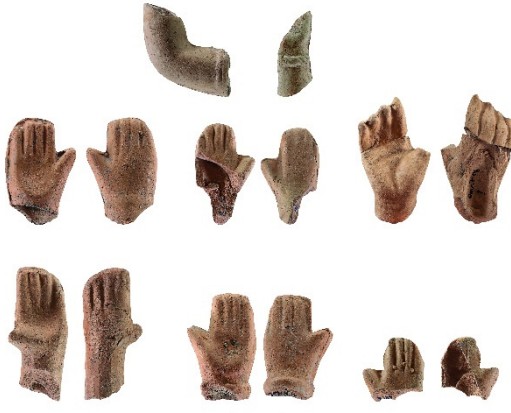

**Figure 5.** Terracottas depicting arms and hands from Omrit (photo by Gaby Laron).

Exceptional is an intact figurine of an arm, handmade and solid, in buff clay, echoing the upraised-arm gesture of this series, and about the same size (Figure 6). This figurine could have served as an archetype to create the plaster molds of this series. This might have been the case if only the production was local; however, as mentioned above, the clay of some 10 fragments was analyzed and points to the Phoenician coast as their origin, some 40 km west of Omrit. However, blocks of clay may have been imported to Omrit, from a distance amounting to a two-day overland journey, as happened sometimes in the production of Hellenistic terracottas (Uhlenbrock 1990, p. 19, n. 19). If so, the molds and terracottas would then have been locally made using this archetype-arm (and other body parts that are now lost). The recurring type of the large-scale group suggests local production was designed to meet the specific needs of the shrine. In Egypt, production centers were traditionally located near shrines (Ballet 2002). This was also the case with the workshops at Corinth in Greece, most of which were situated in the potters' quarter and supplied their goods to the shrines (Merker 2000, pp. 20, 344–46). Thus, the solid arm in Omrit may indicate local production for the needs of the temple, perhaps using blocks of imported clay.

Another possibility is that this limb was an anatomical votive as found in healing shrines (Graham and Draycott 2017; Hughes 2017), but the exceptional nature of the Omrit find, and the bracelet, atypical of anatomical votives, do not indicate this alternative. Yet another possibility is that the solid arm was not an archetype at all, but may have served as a symbolic abbreviation of the type, attesting to the significance of the upraised-arm gesture.

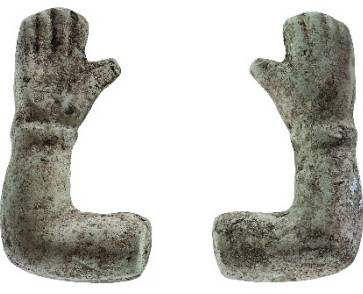

**Figure 6.** Terracotta of a solid arm from Omrit (photo by Gaby Laron).

Torso fragments are also instructive as to the type. Two of them (Figure 7) present the basic type of standing figure with an upraised arm, one is dressed in a sleeveless garment and wears a double-plain bracelet on the arm, whereas the other is nude. The drapery or nudity may indicate gender or age. Another fragment shows a chubby, childish belly with the navel incised.

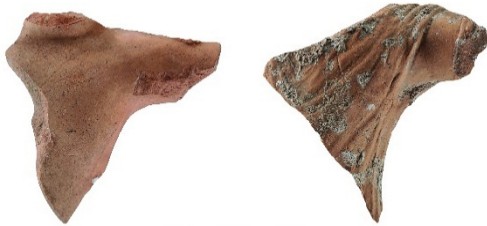

**Figure 7.** Terracottas of torsos with upraised arms from Omrit (photo by Gaby Laron).

Three fragments depict torsos wearing a crescent-shaped pendant (Figure 8) on their chest. This kind of jewelry is well attested in ancient Egypt, the Levant, and Mesopotamia from the Neolithic period onward, and served as an apotropaic amulet in various cultures (Wrede 1975; Golani 2013, pp. 157–58; Ilan 2016). This amulet was also known in Greece and Rome, called in Latin *lunula* (pl. *lunulae*) (Dasen 2003, p. 280; 2015, pp. 189–90). Two of the fragments show a *lunula* on a nude body, whereas another depicts a *lunula* worn by a girl draped in a chiton painted in blue.

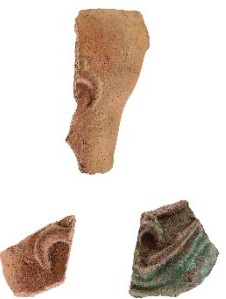

**Figure 8.** Terracottas of torsos wearing a *lunula* pendant from Omrit (photo by Gaby Laron).

Eight fragments depict feet and/or legs (Figure 9). Some of the legs are portrayed barefoot, with the underside curved in a naturalistic manner, or with the toes indicated. Others wear shoes, with signs of shoelaces or grooves decorating the surface.

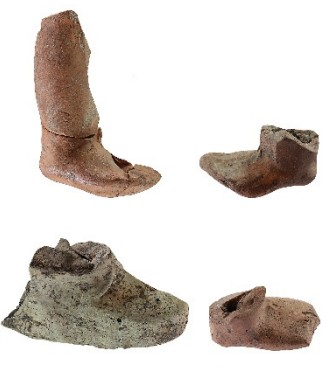

**Figure 9.** Terracottas depicting legs and feet from Omrit (photo by Gaby Laron).

Reconstructing the age type from the various fragments, this group portrays young children, mostly toddlers, whose young age is evident by the chubby faces and bodies of most. They are portrayed standing, according to the pose of the feet fragments. Some are nude, others are draped in a sleeveless tunic; the same is true for the feet—some are portrayed barefoot, and others are shod. A most significant and repeated motif is their arms lifted up from the sides and bent at the elbow, with the palm open and turned outward, as evident from the torso and arm fragments. Although we have only fragments, and hypothetically the arms could be positioned otherwise, the uplifted gesture is the most plausible one. Setting the arms forward with the palms turned down is unlikely, since the detailed open palms are modeled to be seen in a frontal view. Any other pose does not fit the fragments or natural bodily stance. Therefore, the "hands up" pose is most likely, considering the modeling of the limbs, as well as the parallels discussed below. Unfortunately, no fragments of sex organs were detected. The arms, and sometimes the torsos, carry bracelets and pendants (Figure 10).

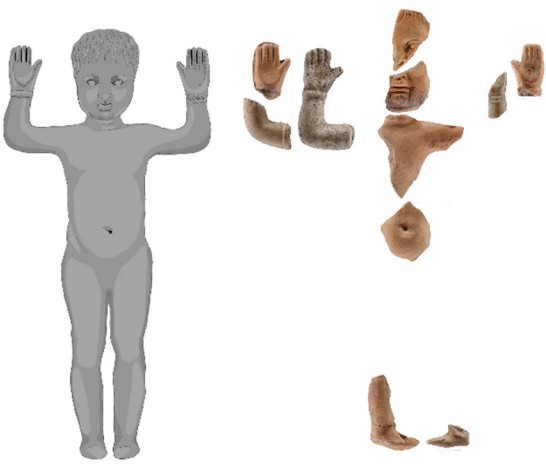

**Figure 10.** Reconstruction of the type (drawing by Zohar Slaney).

Due to the fragmentary state of the terracottas, it is hard to say how many complete figurines of this group are represented in the assemblage. Only a rough estimation can be offered. The assemblage includes 10 different head fragments, approximately 12 right hands or forearms, 9 left hands, 8 feet (both left and right), and numerous body fragments, including nude and draped torsos, shoulders, and limbs, which according to their fabric and size belong to different figurines. Thus, one may estimate about 20 to 40 figurines represented in the corpus.[2]

### 3.1.2. Parallels to the Type

Large-scale figurines are typical of late Hellenistic and Early Roman terracottas from production centers in Asia Minor, for example from Smyrna (Hasselin Rous et al. 2009, pp. 100–99). Large-scale figurines with articulated limbs also appear at sites from the Hellenistic and Roman Levant. A large figurine of a baby was found at Subterranean Complex 169 at Maresha in Israel, dated to the second century BC (Erlich 2019, pp. 229–30). Large-scale babies in terracotta appear also at Hellenistic and Parthian Mesopotamia (Menegazzi 2014, p. 357) and at Greco-Roman Egypt (e.g., Török 1995, pp. 134–35, pls. CII–CV). At Caesarea, a few figurine fragments of the first–second centuries AD depict articulated free-standing statuettes and figurines (Gendelman 2015, pp. 25–30, 33, no. 24). One arm is decorated with a bracelet, and it seems to be raised (Gendelman 2015, p. 30, no. 16).

The best parallels to our group come from the Hellenistic shrine at Kharayeb, near Tyre. This is a small shrine in a rural setting, which operated from the fifth to the first centuries BC. Around the shrine and in its favissa over 1000 terracotta figurines were found, dated to the Persian and Hellenistic periods (Chéhab 1951–1954; Kaoukabani 1973; Oggiano 2015). Among the figurines, a dominant theme is of children: boys and girls from toddlers to adolescents are depicted holding or playing with animals, carrying fruit, learning how to write (*ephebes*), and engaging in other daily life activities (Castiglione 2020). Some children's heads resemble those of Omrit, with their finely modeled faces (Chéhab 1951–1954, pls. 90, 92, no. 3). Similar large-scale limbs were found on the surface of the favissa and therefore are dated to the later phase of the shrine—first century BC (Chéhab 1951–1954, p. 124, pl. XCVII). One of the arms has a bracelet similar to the Omrit figurines. The Kharayeb shrine figurines also resemble our group in the nature of the find spot (a shrine), and in the general popularity of children's imagery in its repertoire. All the examples above are sporadic parallels; we know of nowhere else with a group of this specific and repetitive type, in a temple or elsewhere.

The gesture expressed in the upraised arms with palms turned outward, identifying the figure as "orans" or prayer, is common around the Mediterranean and often interpreted as communication between worshippers and their deities (Leclercq 1936, pp. 2291–97). It is popular in Greek and Roman art in depictions of deities and worshippers alike. Regarding mortals, plenty of visual and literary evidence support the interpretation of this gesture as praying or approaching deities (Neuman 2011, pp. 77–81). The orans gesture appears also in terracotta figurines representing votaries addressing their deities, such as female worshippers from the Thesmophorion at Thasos (Muller 1996, pp. 476–78). Figurines of squatting children of both sexes, in a similar pose, were found in Roman Egypt; in a few cases they are adorned with *lunula* and bracelets, and it was suggested that they represent participants in rites of passage (Török 1995, pp. 127–28, pl. XCIII).

This gesture also appears in the region in Hellenistic and Roman contexts. The Umm El-Amed reliefs from the Hellenistic period show votaries and priests with one arm raised in a similar pose—always the right arm—while the left hand holds cultic vessels, such as the one discussed above (Annan 2013; Michelau 2015). In Roman Palmyra there are altars with depictions of worshippers lifting both hands with the palms turned outward (Ploug 1995, pp. 66, 259, 260; Heyn 2010, p. 632). Closer to Omrit there is a relief on a podium at the temple in Nihaa in the Beqaa, Lebanon, depicting a female figure standing in a similar pose near an altar, accompanied by a man and a winged riding child or god (Krencker and Zschietzschmann 1938, p. 111, fig. 149; Aliqot 2009, fig. 111), or just votive hands made of bronze from the Beqaa (Bel et al. 2012, pp. 218–21). Still closer to Omrit, at Ain Aata on Mount Hermon, a relief was found of a man next to a bull, placed in an arch, raising his right arm in a similar way (Aliqot 2009, fig. 239). Thus, it appears that the orans gesture, with upraised arms or arm and an open palm turned forward, is not strange to the region. The association with children though, is quite unique.

## 4. Discussion: Images of Children in Greek and Roman Sanctuaries

Children are a vital link in society, bridging generations and guaranteeing continuity. In ancient societies, however, children were also the weak link, as their mortality rate was very high, and many did not reach the age of five, due to disease and poor sanitation (on infant mortality and age at death in the Greek and Roman world, see Wiedemann 1989, pp. 17–25; Bradley 2005, p. 69; Carroll 2018, pp. 147–51). Parents were concerned about fertility issues, successful deliveries, survival of newborns and the health and welfare of young children, and consequently they often turned to gods and other superpowers to ensure success and protection (Miller Ammerman 2007; Graham 2014, pp. 36–39; Glinister 2017, pp. 137–39). In Greece and Rome, amulets were attached to babies and children to guarantee their survival, wellbeing, and growth, as shown in archaeological records and visual depictions (Beaumont 2012, pp. 61–63; Dasen 2003, 2015; Carroll 2018, pp. 99–109). Rituals involving children in Greek and Roman families followed stages in early life: birth, teething, weaning, naming, introduction into the family and/or society, etc. (Beaumont 2012; Dasen 2009, 2014; Carroll 2018, pp. 44–47, 63–66).

These parental and social concerns sometimes have visual manifestations in stone and terracotta statuary (Beer 1987; Stucky 1993, pp. 29–39; Miller Ammerman 2007; Terranova 2014; Carroll 2018, pp. 44–50). Such figurines and statues have been found in Greek temples, often depicting crouching children, probably younger than one year (Hadzisteliou-Price 1969; Beer 1987, pp. 23–24; Beaumont 2012, pp. 64–65; Petsalis-Diomidis 2016, pp. 64–67), but in some cases, such as the Artemis compound in Brauron, standing infants and children (both boys and girls) were depicted (Cosi 2001, pls. 12–13). As for the Roman world, in Italy and the northwestern provinces, the most common example is of terracottas of swaddled babies dedicated in temples and graves (Graham 2014; Carroll 2018, pp. 70–81). Various interpretations have been given in the past to the statuary of children in temples; among them, images of gods, companions to the gods or children consecrated to the gods (Hadzisteliou-Price 1969, p. 107). Another view perceives these images as representations of votaries granted to the gods as substitutes for real children (Carroll 2018, pp. 47–50).

Two kinds of cults are often suggested in relation to children's images in temples—the first focuses on concerns for children's health and welfare, and the second involves rites of passage. Some of these figurines and statues are found in healing sanctuaries accompanied by inscriptions, where their healing connotation is clear (Stucky 1993, pp. 134–37; Miller Ammerman 2002, pp. 315–16, 2007; De Cazonve 2008, 2013; Petsalis-Diomidis 2016). These images were given either as a request for healing and welfare (votive) or in payment of a vow (ex-voto) after recovery (for the different types of offerings see Osborne 2004; Derks 2014; Graham and Draycott 2017). However, they are not related solely to a health cult, but to crucial milestones in life, such as teething, un-swaddling a baby, weaning, etc. (De Cazonve 2008; Dasen 2009, 2014; Graham 2013, 2014; Derks 2014).

The two cults, health and rites of passage, were closely related. Teething and weaning were considered vulnerable times for babies, who were exposed to diseases, as attested in Roman sources (Carroll 2018, pp. 64–65), and therefore overcoming them was a proper trigger for a rite of passage. These rites have further significance in terms of socialization, in introducing the infant to the family, to the cult and to society in general (Caneva and Delli Pizzi 2014, pp. 511–14; Graham 2014, pp. 38–39; Glinister 2017).

## 5. Images of Children in the Levant: The Case of Cyprus and Phoenicia

Images of children are often found in a cultic context in the Levant during the second half of the first millennium BC. Most prominent is Cyprus where, from the fifth century BC onward, limestone statues of seated infants, the so-called "temple boys", are numerous (Beer 1987, 1993, 1994; Yon 1994; Caneva and Delli Pizzi 2014). Temple boys also occur on the Levantine coast, in the Phoenician temple of Eshmun near Sidon (see below), and sporadic examples in stone and terracotta from Israel (Stern 2010, p. 17, pl. 11). The type depicts seated children, mostly boys but occasionally also girls, dressed in a tunic, wearing various headdresses, and carrying amulets hung on a belt. The boys sometimes lift their

garment to expose their genitals. They were related to various deities: Melqart, Apollo, Astarte, and Aphrodite. They have been variously interpreted as divine children, crippled children, dedications for a request or thanksgiving for a child, children brought to serve in the temple, prostitution, and rites of passage marking circumcision, teething and weaning (for the different opinions see Beer 1987, pp. 22–23; 1993, pp. 127–35; Caneva and Delli Pizzi 2014).

As for Phoenicia, images of children from that time were found in the Eshmun compound near Sidon and in the shrine of Kharayeb, near Tyre. The sanctuary dedicated to Eshmun in Bostan ech-Sheikh, located near a river about 3 km north of Sidon, is associated with two main aspects—healing and children's welfare (Stucky 1993, 2005; Nitschke 2007, pp. 93–133). The complex operated from the sixth century BC to the Byzantine period, and its main phases were the Persian and Hellenistic periods. According to inscriptions and historical sources, the main god worshipped at the site was the Phoenician god Eshmun, later identified with Asclepius. Dozens of votive statues portraying children were found in the compound, dated from the late fifth century BC to the Roman period (Stucky 1993, pp. 29–39). The statues, some in good shape and others fragmentary, include about 40 crouching children of the Cypriot temple-boy type (including two temple girls), 25 standing children, most of whom are naked boys, a couple of seated boys and girls, and 20 heads. Some of the temple boys are shown holding a toy (astragal) or an animal. The children belong to three distinct age groups: first, the temple boys that represent crawling babies, i.e., less than one year of age; second, standing toddlers around three years old; and third, a smaller group of children close to puberty (Stucky 1993, pp. 38–39).

The Hellenistic phase at the shrine includes a building with friezes in a relief depicting children, adjacent to a room with an empty throne, also known as an "Astarte throne" (Stucky 2005, pp. 147–81; Nitschke 2007, pp. 108–12; Bonnet 2015, pp. 216–17). The frieze is unparalleled to date (Stucky 2005, pp. 172–81, pls. 19–23). The children are portrayed riding deer, holding or playing with animals (dogs, geese, roosters), and other cultic objects (juglets, palm leaves, graffiti-like tripods). The frieze is composed of scenes of daily life (playing with animals) as well as cultic or mythical depictions (riding a deer). The scene evokes associations with Dionysos, Apollo, and Eshmun (Stucky 2005, p. 179), but above all, the details and composition recall processions of Dionysiac rites (Talgam and Weiss 2004, pp. 77–85). Similar to the Omrit group, both boys and girls appear in the frieze; they are draped, half-draped, or nude. Interestingly, some of the figures have one arm raised, and one (Stucky 2005, p. 174, fig. 102, K, pl. 21, no. 2) seems to raise the arm in a pose similar to the Omrit group.

The Eshmun sanctuary provides further testimony of the presence of children in the cult that performed there: about 30,000 glass counters were discovered in the Hellenistic pools at the site; these are thought to have served in Eshmun's healing cult. Dunard (1978) maintained that they were toys given to children to distract them while performing the rituals, and that they were thrown as a game into the niches and holes cut at the edges of the pools (for glass counters as playthings in Phoenicia see Erlich 2017). The evidence for children's involvement in cults in the Eshmun sanctuary near Sidon is therefore solid, and comprises votive statues, a relief depicting children in mythical and daily life scenes, and small finds that may be related to children.

Another Phoenician-Hellenistic shrine with a predominance of children's imagery is that at Kharayeb, mentioned above. The terracottas of children at Kharayeb were interpreted by Chéhab (1951–1954, pp. 125–60) as evidence for agriculture and fertility cults, and initiation into Greco-Egyptian mystery cults. This view was challenged by Lancellotti (2003), who interpreted the cult in Kharayeb as associated with healing and protection of children, similar to the Eshmun sanctuary in Sidon (see also Bonnet 2015, pp. 245–50; Castiglione 2020).

## 6. The Children of Omrit

The terracottas of children from the temple at Omrit fit into a wider phenomenon of children's cults in the Greco-Roman world and specifically in the southern Levant. Since only fragments survived, we must consider the small details in order to reconstruct their use in the local cult. The two main questions are: (1) who are these children depicted in the figurines, and (2) to what kind of cult do they attest?

One may argue that the figurines are symbolic depictions rather than accurate models, but the heterogeneity and individualistic appearance of the figurines point to a realistic depiction. Within the rather uniform type and gesture of a child standing and raising arms, the variety is striking.

Each figurine seems to have been cast from a different mold; some children are dressed while others are naked, some wear shoes and some do not, some are with *lunula*, some without, and their faces—-while somewhat resembling each other—are not alike. Only the hands with open palms are similar, albeit not identical. This individuality is odd considering the technology of production: these are not stone or marble statues each carved separately, as were the children's images from Greece, Cyprus, and Phoenicia. Rather, they are terracotta figurines, cast in molds that can produce multiple copies. Yet, each figurine is different. Although among Hellenistic assemblages of terracottas it is extremely common that figurines are individualized, figurines from Hellenistic and Roman temples in the Levant present serial production of figurines. Such is the case at Kharayeb in Phoenicia, where up to 10 figurines were made out of one mold (Chéhab 1951–1954, p. 120), or at Amathus in Cyprus, where figurines come from serial production (Queyrel 1988). The multi-mold technique, using separate molds for the head, torso, and limbs, provided flexibility and mixture of individual and conventional components. The variety of figures in these figurines implies that they represented individual votaries. It seems that a mass-production method was employed to produce one-time and tailor-made products. The work invested in each figurine indicates it is individualistic and specific, rather than a generic reproduction cast over and over again.

*Age and Gender*

Different sizes of figurines portraying swaddled babies were considered by De Cazonve (2008, pp. 276–77) as signs of different ages, while Graham (2014, p. 33) associated variability in size to economic factors, since size affected the price. The height of the Omrit children, as noted, varies from ca. 30 to ca. 50 cm, and that may indeed have correlated to the child's age. At any rate, although the height of the figurines is not realistic, they are models of real children.

Besides size, which is an uncertain indication of age, some physical elements help reconstruct age, despite the fragmentary condition of the figurines. The foot fragments point to a standing pose, suggesting an age of more than one year, when toddlers can already stand. The chubby faces and in one case a swollen belly as well, suggest any age of less than five years. According to the standing posture and raised hands (or one hand) in an orans pose, any age above two years is reasonable, as younger children cannot be disciplined to stand still in such a pose. One head seems to be that of an older child, perhaps just before puberty (see above, Figure 4, upper left).

Another means of deciphering age is whether the children are clothed or nude. The act of dressing and undressing is an important part of marking stages of life and socialization, such as the swaddling of babies, the first and second toga a Roman boy would have received to mark his changing status through life, a bridal dress and shrouds (Graham 2014, pp. 39–41; Glinister 2017, p. 145). The variety of nude and dressed bodies of the Omrit children may point to different stages in early life. The Cypriot temple boys are mostly dressed, with some exposing their genitalia (Beer 1994, pp. 8–15). In contrast, at the temple of Eshmun near Sidon, the crouching toddlers are naked, whereas the standing children are either nude, half-nude or draped (Stucky 1993, p. 29), what may indicate different

ages. Likewise, it may be assumed that the nude toddlers at Omrit are younger than the clothed ones.

To sum up the matter of the age of the Omrit children: they seem to be between two and ten years old, but most of them range between two and four years old. This span corresponds with some of the statues of standing children in the Eshmun temple near Sidon (Stucky 1993, p. 37). It must be noted, however, that at the Eshmun temple younger, crouching children are the majority, while they are lacking in the Omrit temple. The children portrayed in the numerous terracottas from Kharayeb also varied in age and iconography (Castiglione 2020), unlike the homogeneous group of Omrit. This renders the cult at Omrit more specific and defined than that held near Tyre and Sidon.

It is hard to decipher the gender among the children portrayed in the Omrit figurines. Some heads seem male, but certain pieces showing cloth and accessories point to girls. Most important in this regard are the three figures wearing a *lunula* pendant hung on a string, one of which is also dressed in a feminine chiton painted in blue (see above, Figure 8, bottom right). Across the Mediterranean in different periods the *lunula* was associated with women and children of both sexes, more rarely with men and even with animals (Wrede 1975). In the Roman world the *lunula* was known as an amulet for girls, parallel to the bulla worn by boys (Olson 2008, pp. 144–45; Glinister 2017, pp. 141–45). This amulet for girls was related to the moon and to the monthly menstrual cycle (Dasen 2015, pp. 189–90). Unlike the bulla, common in Roman portrayals of boys, the *lunula* is quite rare in archaeological record and visual representations (Dasen 2003; Glinister 2017, pp. 141–42; Carroll 2018, pp. 48–50). On children's tombstones for example, the bulla is common while the *lunula* absent (Mander 2013, pp. 74–75). *Lunulae* are depicted on a girl on the Ara Pacis altar (Olson 2008, p. 145, fig. 6.2) and also occasionally on terracotta figurines of girls (Grandjouan 1961, pl. 10, no. 458). In Egypt and the Levant, *lunulae* are more common for both sexes, but mostly females. They appear on Roman mummies from Egypt (Walker and Bierbrier 1997, pp. 164–65) and rarely on the Cypriot temple boys (Beer 1993, p. 28).

Why do girls wear only *lunula* in the Omrit figurines, whereas bullae or other amuletic pendants are missing? Some of the figurines may have been adorned with real bullae—-leather-made and hung on a string—-while *lunulae*, which were made of gold and were precious, were only depicted. This lack of *lunulae* at Omrit recalls the scarcity of real lunulae in tombs as well, probably because they were passed down to living members of the family due to their costliness (Dasen 2003, p. 288).

The *lunula* may also have served to differentiate between girls and boys, since sometimes it is hard to tell girls from boys when they are toddlers. Moreover, perhaps the *lunula* was not a gender indicator at all, but was worn by both genders. At any rate, it is clear that the Omrit group is composed of boys and girls alike. The preponderance of boys so evident in the Cypriot and Sidonian statues (Stucky 1993, p. 29; Caneva and Delli Pizzi 2014, pp. 506–8) does not occur at Omrit.

Two prominent features that might further clarify type and seem to characterize all the figurines in the series, are the amulets and the pose. All the arm and hand fragments show the same gesture of upraised arm with open palm turned outward, and they all wear bracelets. This gesture is generally associated with prayer or benediction, as mentioned above. When performed by adults, the gesture is sometimes related to the initiation of a child into the Dionysiac mystery cults (Talgam and Weiss 2004, p. 85). Noteworthy in this regard is the fact that initiation rites are considered the archetypical rite of passage (Mouton and Patrier 2014, p. 4). Notwithstanding, the figures at Omrit seem to have both arms raised, and by the children themselves.

Bracelets and *lunulae* are protective amulets that were given to children to shield against evil spirits until the they came of age (Dasen 2003; Bradley 2005, pp. 89–90). Amulets are commonly depicted on statues and terracottas of babies and children in cultic contexts around the Mediterranean, e.g., toddlers' images on Attic choes (Carroll 2018, p. 47), swaddled babies from Italy and Gaul (Dasen 2003, 2014, pp. 242–45; Miller Ammerman 2007, pp. 142–45; Graham 2014, pp. 30–31) and the Cypriot temple boys

(Beer 1993, pp. 18–31). The children from the Eshmun temple near Sidon carry no visible amulets, although Beer (1993, p. 91) suggests that they could have been painted on the statues. Furthermore, since most of the fragments depict nude bodies, but all arms carry bracelets, it seems that even nude children carried them. This stresses further their cultic and apotropaic association, not being mere fashionable accessories. The single solid arm with bracelet (see Figure 6, above), if not an archetype for molds, may have served as an abbreviated symbol of the type.

To sum up, the fragmentary figurines from Omrit portray children of both sexes, mostly around two or three years old, wearing amulets and raising their arm or arms in a ritualistic gesture. The type is specific and consistent, but with a great deal of individualism. These votives were probably brought to the temple by worshipers—-the parents and their children whom they represent, imitate, or perhaps replace.

### 7. Rites of Passage at the Omrit Temple

As noted, the two most common circumstances for children's ceremonies are healing and rites of passage, and the two are often related. Healing rites are suggested in the Phoenician cult centers, at the Eshmun temple near Sidon, where the connection to the healing god has been well established (Stucky 1993), and also in the shrine of Kharayeb near Tyre (Lancellotti 2003). At Omrit, the proximity of the temple to a water source may also associate it with healing (Derks 2014, p. 59). However, the repetitive type of figurines, their specific gesture, and the lack of babies or very young infants, point to a more specific cult.

I would suggest that these figurines are related to rites of passage, in the transition from infancy to childhood, and perhaps also at other stages, for example, puberty. The statues at the Eshmun temple near Sidon depict children of various ages at different stages of childhood (Stucky 1993, pp. 37–39). At Omrit, however, most of the fragments belong to toddlers, while only one looks older. In the analysis below, therefore, I concentrate on early childhood.

Children between two and four years of age are at a crucial threshold in life. This is usually the final stages of weaning and change of diet, and from the physical perspective, children of that age develop their motor skills and improve mobility. From the mental perspective, this is the time when children learn how to talk and communicate and are more easily educated and socialized. These physical and mental stages mark the transition from baby to child, and it is reasonable to believe that they involved ceremonies (Derks 2014, p. 64). Scholars believe that similar circumstances were associated with the dedication of temple boys in Cyprus (Beer 1993, pp. 134–35; Caneva and Delli Pizzi 2014, pp. 510–11), and with younger children at the Eshmun temple near Sidon (Stucky 1993, pp. 37–38).

Today, the age of three is still considered a watershed in early life, and the beginning of socializing of children to both family and community; for example this is the recommended age for pre-school education in the OECD countries.[3] In certain ultra-Orthodox Jewish communities, boys have their first haircut at three years old, and that is the age a boy is transferred from the women's to the men's domain; he wears his first *tzitzit* (ritual fringe) and *kippah* (ritual cap), similar to the first and second togas, *toga praetexta* on early childhood and *toga virilis* on adolescence, given to ancient Roman children of a certain class to mark milestones in life and introduction to society (Rawson 2003, pp. 142–45; Edmondson 2008, pp. 26–27). Girls in these communities are considered young women in terms of modesty laws and dress code at three years old, and both genders start studying the alphabet at pre-school at this age. Some of these changes are marked by ceremonies, such as the first haircut for boys (called *halakeh* or *upsherin*), celebrated with the family at a special high place, the sacred tomb of Rabbi Shimon Bar Yochai in Meron, Galilee (on the first haircut and other ceremonies for Jewish boys, see Bilu 2003). This annual celebration at Meron is today the largest Jewish festival in the world, drawing over 200,000 people every year, of which a small part is the celebration of the first haircut for boys around three years old.[4]

Some 40 km away and 2000 years apart, similar ceremonies may have taken place at the temple of Omrit. Above all, they involved the dedication of images of the children

undergoing these rites of passage. That may explain the individualism we see in the otherwise rather uniform type of figurines from Omrit.

Let us try to reconstruct the ritual at the temple, in light of the above, although we have little evidence for the rite itself. Families would have gathered at the temple from nearby settlements. There may have been a ceremonial procession of families, involving nude and draped children, similar to the Dionysiac procession depicted in the mosaic from the House of Dionysos in Sepphoris (Talgam and Weiss 2004, pp. 77–85, color pls. 8–10). This may have taken place during a festival or on individual occasion. The figurines portraying the children were probably commissioned in advance to fit the age and gender of the child or other preference of the family. Otherwise, shops in or near the temenos would have offered a variety of figurines; one could have chosen a girl or a boy, draped or clothed, with or without *lunula*, shod or barefoot, babyish in appearance or older, etc. This procedure resembles a modern custom in Cyprus, where babies modeled in wax are prepared individually and sold to parents, who dedicate them in churches where they are kept hung on the walls (Miller Ammerman 2007; Graham and Carroll 2018).

The cult may have taken place in a dark location or at night, if the numerous oil lamps found in the shrine are related to these rites (Rosenthal-Heginbottom 2017, pp. 453–54). During the time a ritual was performed at the temple, it probably involved raising the arm or arms. One can imagine action songs sung by a priest or the parents to urge the child to raise his or her arms, as in the popular children's song "Head and shoulders, knees and toes", or the child may have been given verbal instructions. Perhaps the ceremony involved crossing a real threshold, that of the temple, as rites of passage rites occasionally involve crossing a threshold to signify a linear, no-return change of status (Mouton and Patrier 2014, p. 8; Graham 2014, pp. 38–39). The child must have gone through some physical change and manipulation of the body, such as the abovementioned raising of the hands—-dressing, undressing, putting on or taking off amulets, a haircut, etc. Sacrifices were made, prayers chanted, gifts given; and then the child's image was left in the temple, while the real child, now initiated into his or her new self, returned home with his/her family.

Similar steps are recorded in van Gennep's study on rites of passage: the first haircut in Hindu practice (van Gennep 1960, pp. 54–55), active participation of the child and a wooden image of him serving as a substitute in China (van Gennep 1960, pp. 55–58) and more. Demolishing the sacred object while passing through the doorway and crossing boundaries in childhood ceremonies is common in traditional societies, or in van Gennep's words (1960, p. 60):

> "The destruction of the object used in the rite may be explained by the fact, observed in Australia, South America, and elsewhere, that sacra may be used only a single time; as soon as a ceremonial phase is ended, they must be destroyed (this being the central idea of sacrifice) or put aside, as if emptied of their powers, and for each new phase there must be new sacra, such as new bodily ornamentation, costumes, or verbal rites".

This can explain the fragmentary state of the figurines at Omrit, scattered around the temple.

The rather large scale of the figurines, compared to the average size of figurines in the Levant, is also worthwhile of discussion in regard to rites of passage. Small-scale figurines were often discussed in relation to miniaturization theory, suggesting that they evoke physical proximity, emotional connection, and intimacy with their users (Bailey 2005, p. 34; Langin-Hooper 2020, pp. 24–51). Stewart (1993, pp. 37–69) views the miniature as a metaphor for the interiority of the self. The Omrit figurines are not too big to hold and carry, but not small enough to grasp and negotiate intimately with. We can therefore suggest that choosing to portray the children's image in a relatively large scale was meant to achieve the opposite: to evoke defamiliarization and remoteness, and thus to symbolize the separation from one stage to another in a passage rite (Leach 1976, p. 77). Demolishing the figurine after the ceremony could have served the same purpose.

As mentioned above, the temple at Omrit was an extra-urban temple. We should consider it a boundary temple, set in the rural area of the Hula Valley. In Italy, swaddled babies are typical of boundary temples, such as at the sanctuary at Vulci on the border between city and countryside, or at Marica, where there is a natural border between forest, land, and sea (Graham 2014, pp. 41–42). The two cult centers with numerous terracottas or statues of children in Phoenicia are also out of the city, in the countryside: Kharayeb between Tyre and Sidon (Oggiano 2015), and the temple of Eshmun some 3 km north of Sidon (Nitschke 2007, pp. 106–7). Likewise, the Omrit temple lies on the border of the Golan Heights and the Hula Valley. It seems that cults involving children were conducted in the countryside rather than in the city, whether in relation to healing, rites of passage or any other cult.

The liminal location of such temples fits the transition between milestones of life, reflecting passages over boundaries, both physical and conceptual (Mouton and Patrier 2014; Graham 2014, pp. 41–42). Alternatively, the location of the temples might have been in relation to the personal and familial character of the cult, avoiding the political identification typical of city cults. For example, the rural shrine of Kharayeb, situated halfway between Tyre and Sidon, may have served families of these two rival cities.

## 8. Conclusions

The figurines of the large-scale series discussed here supplement the story of the Omrit temple and render it as an extra-urban temple serving families. We should bear in mind that temples were not necessarily assigned to one cult only, and they may have been used differently throughout the year, during different festivals and to worship various deities. Unfortunately, we have only little evidence on deities venerated at the temple. The large-scale series of figurines represents one part of a cult that may have taken place at the temple.

The large-scale figurines from the Omrit temple are both common and unique. They are common in the sense that they fit into the larger picture across the Mediterranean of images of children in various media, sizes, iconographies, ages, and styles, dedicated in cult places. In contrast, their uniqueness lies in the fact that nowhere else has such a group of children's figurines been discovered with specific gestures and attributes—-upraised arms bearing bracelets. Yet, within this specific iconography, and despite the mold technique that allowed for mass production, each figurine is individual, and seems to represent a specific child. The mixture of generic and individual traits is what makes the Omrit figurines special and allows for the identification and reconstruction of the cult involving them.

The cult associated with these figurines at Omrit probably included rites of passage for young children in the transition from infancy to childhood, around three years of age, as well as other ages. The cults could also have included healing rites, either as a petition for a cure or as payment of a vow by offering the figurines and other gifts to the savior deity. The cult was personal and initiated by families, probably with the aid of priests. It could have taken place throughout the year or on special occasions and festivals.

The Omrit children fit in with the nearby Phoenician temples in the region of Tyre and Sidon, where numerous figurines and statues of children were consecrated. These statues were also offered as part of rites of passage and healing. Rites of passage are evident in Hellenistic Phoenicia, as has been suggested regarding a hoard of a young bride dedicating her personal belonging of childhood to symbolize her upcoming new status and to guarantee a successful marriage (Erlich 2017). However, as much as the Omrit figurines fit the Phoenician realm, at the same time they are very Romanized. The style, fashion, and even their rather large scale, are all Roman, and could have fit in anywhere else in the Roman Empire.

The figurines from Omrit are fragmentary and seem marginal compared to the impressive architecture and decoration of the temples at the site. However, as small as they are, their contribution to the understanding of the cult in the temple is great: they are key clues to the people who made them, bought them, used them in the cult, and finally discarded

them. They allow us to focus not only on the building itself, or on the deities, as is normally the practice in the study of Hellenistic and Roman cult places in the Levant, but on the rituals and the people who performed them.

**Funding:** This research received no external funding.

**Acknowledgments:** I am grateful to the excavators of the temple of Omrit, Andrew J. Overman, Daniel N. Schowalter, and Michael C. Nelson, who entrusted me with the publication of the figurines, and for their valuable comments on the draft of this paper and early versions. I would like to thank Gaby Laron for the photography, Zohar Slaney for the drawing, Silvia Krapiwko for the graphics, Miriam Feinberg Vamosh for the editing, and Henrike Michelau for kindly permitting me to use her drawing. Petrographic analysis was conducted by Anastasia Shapiro of the Israel Antiquities Authority, to whom I am grateful.

**Conflicts of Interest:** The author declares no conflict of interest.

## Notes

1. The site was excavated by A.J. Overman, D.N. Schowalter, and M.C. Nelson (Macalister College excavations) from 2001 to 2011. The temple, its architecture, stratigraphy and some of the finds have been published (Overman and Schowalter 2011; Nelson 2015; Overman et al. 2021). The finds from the compound are being prepared for publication by the excavators and a team of scholars. I am grateful to the excavators for entrusting the terracottas assemblage to me.

2. In the recent excavations at Omrit, conducted by Jennifer Gates-Foster, Dan Schowalter, Michael Nelson, Jason Schlude, and Ben Rubin, in the settlement outside the temple compound, more fragments of this type were found. They are currently being studied by the author.

3. https://www.oecd.org/els/soc/PF3_2_Enrolment_childcare_preschool.pdf (accessed on 11 April 2022).

4. Recently a disaster occurred at this celebration, see: https://en.wikipedia.org/wiki/2021_Meron_crowd_crush (accessed on 11 April 2022).

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
