# Peer review of "Children in Need: Evidence for a Children’s Cult from the Roman Temple of Omrit in Northern Israel"

_religions, doi:10.3390/rel13040362_

Round 1

Reviewer 1 Report

The article deals with the topic of childhood examined through particular terracotta figurines discovered in the ancient site of Omrit. It is very well structured, with an updated bibliography, and it presents new data, compared with other documentation known from different sites, especially from the eastern Mediterranean. The paper offers new food for thought for the coroplastic studies, the knowledge of the Hellenistic and Roman local communities, as well as for the rituals and cultural habits in this part of the Levant, and for a deeper understanding of the childhood from an archaeological and anthropological perspective.

I have only one suggestion: if it is possible, I propose to insert a figure 3 bigger, since the details discussed in the article are not clearly visible in the current size.

I indicate a few typos:

  • line 85: Erlich 221,144–53 = Erlich 2021, 144–53
  • line 340: Catiglione = Castiglione
  • line 398: (Castiglione Forthcoming) = (Castiglione 2020)
  • line 609: (Castiglioni 2020) Castiglioni,  = (Castiglione 2020) Castiglione
  • line 611: Chéhab, Maruice = Chéhab, Maurice 
  • line 612: Cosi, 2001. L’arketia di Brauron e I culti femminili. Bolgna: n.p. = Cosi, Dario M. 2001. L’arketia di Brauron e I culti femminili. Bologna: n.p.
  • line 652: Carroll, Marueen = Carroll, Maureen
  • line 665: Kaoukabani, I. = the name is in the short form
  • line 704: Yehudit 2011). = Yehudit 2011. please, cancel the bracket
  • line 706: Michael C. (eds.). = Michael C. (eds.) 2021.
  • line 718: Daniel N. = Daniel N. 2011.
  • line 731: Talgam, Rina and Ze’ev Weiss. 2004 = please, check the names
  • line 733: La presenzadei bambini = La presenza dei bambini
  • line 741: Wrede, Henning. = Wrede, Henning 1975.

Finally, some points regarding the page layout and the editorial revision:

  • Please, check the left margins in:
    • line 611: (Chéhab 1951–54)
    • line 679: (Miller Ammerman 2007)
    • line 700: (Olson 2008)
    • line 706: (Overman, Schowalter and Nelson 2021) 
  • There are more than one space in line 613: revue d’études.

Author Response

Thank you for your kind words.

I have corrected all the typos except for:

Fig. 3 - I hope the publisher can enlarge the figure.

Line 665: Kaoukabani, I. = the name is in the short form - Sorry, I could not find the name. 

Please see the attachment of the revised paper according to you suggestions.

Reviewer 2 Report

Overall, this article is presents an important corpus of archaeological finds, namely a group of fragmentary, relatively large-scale figurines of children from a sanctuary context. The author gives the background of the excavation and the sanctuary site, an overview of the figurine corpus, and some hypotheses regarding their use in coming of age rituals. The methodology is sound, and fits very much within standard archaeological approaches to studying figurines. It is solid, although not terribly innovative. However, there are some points of analysis which require further work before the article should be published. There are several points of iconography on which the author jumps to conclusions, such as on the ritual nature of the upraised arm gesture and the amuletic function of the bracelets. These are possibilities, but they are not sureties, and I was not convinced with the author's arguments about them as they currently stand. The author also needs to engage much more substantially with current theoretical research on the archaeology of childhood, and the role of children in ancient societies - this is particularly needed in the concluding sections of the article. In reaching towards a specific conclusion about the figurines' role in coming of age ceremonies, the author has also ignored or overlooked other possibilities, such as connecting these larger-scale figurines with the smaller-scale depictions of Aphrodite and Herakles that come from the same temple. A more site-specific conclusion, rather than a generalized and imaginative reconstruction, would have been more beneficial to the reader. Overall, this is an important study - the material is very interesting - but the author could have done more to analyze it in a way that will be useful for other scholars.

Specific notes:

Page 3, line 85: the Erlich citation has a typo in the year

Page 6, lines 186-187: "A most significant and repeated motif is their arms lifted up from the sides and bent at the elbow, as in modern gesture of surrender, with the palm open and turned outward." I'm not sure that this connection to the modern meaning of the gesture is helpful here, as it plants an idea in the reader's mind which may have little to do with the ancient meaning. 

Page 7, Figure 10: The genital region of this reconstructed figure is left ambiguous. I assume this is to avoid deliberately sexing the figure - and that this is an artistic convention of the illustrator, and not actually how these figurines would have appeared? Would they have had genitals? There should be some discussion in the text about why the figure is reconstructed in this way.

Page 7, section 3.1.2 Parallels to the Type: There are also large terra-cotta "babies" from Seleucid/Parthian Mesopotamia (at Borsippa, Babylon, Seleucia-on-the-Tigris) as well as Ptolemaic/Roman Egypt. These comparative examples should also be cited here.

Page 8: Paragraph on the meaning of the gesture (lines 223-238). This section needs significant elaboration. It is not convincing as it currently stands. The assertion that this is an "adoration gesture" that was "common around the Mediterranean and often interpreted as communication between worshippers and their deities" needs citations. The comparative examples are helpful, however almost all of them are 1. reliefs of a cultic scene (so, two-dimensional and narrative, rather than three-dimensional and non-narrative, as the figurines are), and 2. depictions of adults. No mention is made of these difference and how they impact the interpretation of the figurines. Indeed, as the author has mentioned, but largely glossed over, none of these figurines are found intact - so how can we be sure that the arms were raised, rather than held outwards or positioned downwards at the sides? Even if the arms were raised, this might mean something quite different on the body of a child, who might be reaching for his mother, divine protection, or even imitating the natural posture of infants lying on their backs, rather than just echoing the ritual behavior of adults. The author discusses none of these possibilities, and instead ends this very brief discussion with the very inconclusive and unsatisfying summary on lines 237-238: "Still, the upraised arm (or more probably both arms) is too general and standard a gesture to be instructive as to its meaning in the Omrit figurines." The author is missing an opportunity here to contribute some meaningful interpretation to these figurines (and it also made me wonder about the utility of this article, if no interpretation will be offered). 

Page 8, line 241: typo in word "children"

Page 8, line 246: "superpowers" seems an odd word choice. Maybe "supernatural forces"?

Page 10, lines 352-364: Although I appreciate the author's attention to detail here regarding the individuality of the figurines and mold use, I question his/her conclusions about the significance of this finding. Among the Hellenistic figurine repertoires in the Ancient Near East, it is extremely common that figurines are individualized, and it can actually be rare to find two figurines made from the same mold. This holds true across motifs - it is certainly not restricted to figurines of children. I'm not sure that I've ever heard a good explanation for why this is - certainly, molds lend themselves to mass production, and that is how they are often used at workshops near Greek sanctuaries (for instance). But that's not how molds were used everywhere. So I am afraid that a figurine's uniqueness (or lack of exact duplicates) is not enough evidence to surmise that these were, in fact, individual likenesses of specific children. "One-time and tailor-made"? Yes, definitely. But something akin to portraits of individual votaries? I'm not sure we have enough evidence.

Page 10, lines 371-372: "A life-size terracotta statue would have been too much work to produce and would have been too fragile and unsuitable to be moved around." Why? There are life-size terracotta statues on Cyprus and from Etruria. If people want them, they can be made. Miniaturization here is a choice, not a necessity - and the author's analysis would be more productive if he/she analyzed it as such. What is the advantage (not just in practical terms, but in affective properties and object experience) in making an already small person (a child) even smaller? There is an enormous amount of theoretical work on miniaturization in figurines (look for Doug Bailey, Rosemary Joyce, Stephanie Langin-Hooper) that could really help the author's analysis of scale.

Page 11: the discussion of lunula and bulla is somewhat repetitive, after already discussing these motifs and gender on page 6. I would suggest consolidating to a single discussion.

Page 12, lines 433-442: the author's assertion that the bracelets worn by the children must be amulets is not very convincing. The author does not point to any specific iconography or images on the bracelets that would make them amuletic. From the images included in the article, the bracelets appear to me as plain bangles, which, if seen on a woman's wrist, would undoubtedly just be interpreted as decorative jewelry. So, why does the author consider them to be amuletic? Is it only because they are on the wrists of a child? If so, the logic is too circular here. 

Page 12, lines 471-continuing on Page 13: discussion of Orthodox Jewish rites of transition at age 3. While this is an interesting possible parallel, the author spends a considerable amount of time on this modern cultural practice. Less time on this, and more time on such practices in the ancient world, would have been more helpful. For instance, the author mentions the ancient Roman practice of giving togas to children at a certain age - but does not elaborate further, even though this would have been useful information.

Page 13: The reconstruction of the cultic ceremony was imaginative, and a useful way of bringing the archaeological data to life for the reader. The hypothesis on lines 513-514, that the "child’s image was left in the temple, while the real child, now initiated into his or her new self" could have used some additional theoretical support/citations about substitute bodies in art. 

Page 13, starting on line 515: the information about the geographic situation of this temple in the hinterland would have been useful at a much earlier point in the article. 

Pages 13-14: the suggestion that infants were seen as not yet being part of society and thus should stay away from "civilized city life" was completely unconvincing. Are there no babies born in cities? The author would have benefitted from much more engagement - here and throughout the article - with research on the archaeology of childhood, which posits that children were actually far more "seen" in the adult spaces of the ancient world than they are today (where we separate children into specific locales, such as schools, playgrounds, children's museums, etc and expect not to see them at their parents' workplaces, etc.). Look at the work of Traci Ardren, Jane Eva Baxter, and Lesley Beaumont. 

Page 14, Conclusions: Why is the reader just now learning that figurines of Aphrodite and Heracles are also found in this temple, and that Aphrodite is also mentioned in an inscription? Why does the author not include this information in the interpretation of the child figurines? Perhaps they are not related, but perhaps they are. These are two of the most popular deities in the Greco-Roman Near East, and they both have important connections with childhood: Aphrodite as mother of Eros, and often depicted with him as a mother, and Herakles as a "baby hero" in the episode of strangling the snakes and saving his own life, as well as the life of his bosom-brother. Angela Heap has even written on "the baby as hero" in Hellenistic drama. I would suggest that such possibilities at least should have been considered in the author's interpretation of the child terracottas.

Author Response

Thank you very much for your through review and useful comments. Please see the attached file.

Round 2

Reviewer 2 Report

I am satisfied with the author's revisions. I think more could have been done to address the literature on the archaeology of childhood, but most of my points were addressed, and I think the article is now a sufficient contribution to scholarship that it deserves to be published.